# Design and Fabrication of Sodium Alginate/Carboxymethyl Cellulose Sodium Blend Hydrogel for Artificial Skin

**DOI:** 10.3390/gels7030115

**Published:** 2021-08-09

**Authors:** Kun Zhang, Yanen Wang, Qinghua Wei, Xinpei Li, Ying Guo, Shan Zhang

**Affiliations:** 1Industry Engineering Department, School of Mechanical Engineering, Northwestern Polytechnical University, Xi’an 710072, China; npu_zk@sina.com (K.Z.); lixinpei@mail.nwpu.edu.cn (X.L.); guoying0402@sina.com (Y.G.); zhangshanzs33@163.com (S.Z.); 2Institute of Medical Research, Northwestern Polytechnical University, Xi’an 710072, China

**Keywords:** SA/CMC-Na, hydrogel membrane, artificial skin, 3D printing, skin regeneration, wound healing

## Abstract

Tissue-engineered skin grafts have long been considered to be the most effective treatment for large skin defects. Especially with the advent of 3D printing technology, the manufacture of artificial skin scaffold with complex shape and structure is becoming more convenient. However, the matrix material used as the bio-ink for 3D printing artificial skin is still a challenge. To address this issue, sodium alginate (SA)/carboxymethyl cellulose (CMC-Na) blend hydrogel was proposed to be the bio-ink for artificial skin fabrication, and SA/CMC-Na (SC) composite hydrogels at different compositions were investigated in terms of morphology, thermal properties, mechanical properties, and biological properties, so as to screen out the optimal composition ratio of SC for 3D printing artificial skin. Moreover, the designed SC composite hydrogel skin membranes were used for rabbit wound defeat repairing to evaluate the repair effect. Results show that SC4:1 blend hydrogel possesses the best mechanical properties, good moisturizing ability, proper degradation rate, and good biocompatibility, which is most suitable for 3D printing artificial skin. This research provides a process guidance for the design and fabrication of SA/CMC-Na composite artificial skin.

## 1. Introduction

Skin defect resulting from burn injury is a major public health issue. Each year millions of people around the world suffer from burn injuries, with nearly 300,000 succumbing to mortality [1]. Surgery or skin grafts are always required for the patients with deep burns, and some patients may die from the infections that are left untreated [2,3,4]. There is no widely accepted ideal skin therapy for severe burn injuries, which is still a challenge in clinic. [5,6,7]. To date, autologous skin grafting following early excision of necrotic tissue has been widely used as the best approach for skin repair and reconstruction [8]. However, autologous skin is usually insufficient for patients with large-area burns due to the lack of donor skin, and allogenic or heterogeneous skin would lead to a strong immune response [9]. With the development of tissue engineering, tissue-engineered skin grafting is considered to be an effective alternative treatment for severe burn injuries with large-area skin defects [10]. Previous research showed that by culturing the patient’s epidermal cells or fibroblasts in vitro and transplanting the cultured cells in the tissue-engineered skin, the native-like skin of the burn patient can be finally regenerated [11]. However, usually the tissue engineering culture period is too long, during which the wound infection can easily result from improper protection [12]. Moreover, whether it is autologous skin grafting or tissue-engineered skin grafting, they are generally subject to some necessary constraints, such as painful skin suture operation and suture excision, which would bring great pain to patients with skin injury [13,14]. Meanwhile, it is difficult to make tissue-engineered skin according to the shape and structural characteristics of the wound with the traditional tissue engineering technologies.

Currently, with the emergence of 3D bioprinting technology, it would be feasible to solve the problem of limited skin donation for large-area skin defects [15]. The 3D printing technology can be used to fabricate components with highly flexible customized structures. Among all the 3D printing technologies, extrusion 3D printing is the most widely used technique to fabricate planar artificial skins [16,17,18,19,20,21,22,23], and this printing process possesses the advantages of being ready to access, easy to manufacture complex structures, and adjustable to print biomaterials [16,17,18,19]. This technology uses pressure to extrude a mixture of biomaterials, and the mixtures are deposited layer by layer to build a 3D structure component [17,18] through a controlled movement nozzle.

Sodium alginate (SA), gelatin, chitosan, and carboxymethyl cellulose (CMC-Na), due to their good biocompatibility and forming characteristics, are widely used in 3D printing tissue-engineered scaffolds [20,21,22,23,24,25,26,27,28]. Among them, water solubility of SA is beneficial to the absorption of wound secretion and the maintenance of a moist wound environment. As well, due to the hemostasis, the porous alginate non-adhesion wound dressings and secondary dressings are commonly used in the treatment of wound and burned skin injuries. In addition, SA has the advantages of low cost and abundance in source, good biocompatibility, suitable viscosity, fast gelling speed, etc., and its gelation is formed almost immediately by calcium ion cross-linking at room temperature. However, due to the lack of arginine-glycine-aspartic acid (RGD) motifs, SA is not conductive to cell attachment [29]. Additionally, the poor tensile stress and compressive stress of pure alginate also greatly restrict its further application as tissue-engineered materials [30]. Therefore, to address the defects mentioned above, some other biomaterials (e.g., nanoparticles, collagen, gelation, and cellulose) are usually used to improve the biological and mechanical properties of SA, to meet the requirements of tissue engineering.

In recent years, a large amount of research on SA has been performed. In the study of Song et al. [30], a hydrogel was fabricated from an aqueous solution of gelatin, sodium alginate, and carboxymethyl cellulose sodium by radiation-induced cross-linking at room temperature. The results suggested that the compressive strength of this hydrogel increased significantly with the increase of sodium alginate content. Du et al. [31] fabricated tubular hydrogel scaffolds of nano-hydroxyapatite/alginate via a layer-by-layer technique. By changing the nano-hydroxyapatite concentration, the elastic modulus of the hydrogel scaffold was adjusted from 0.98 MPa to 2.78 MPa. Choe et al. [20] prepared nanocomposite scaffolds of alginate/gelatin/graphene oxide and applied them for bone healing. The compressive strength of the fabricated scaffold increased 46.7% after the addition of graphene oxide. Chen et al. [32] developed a composite membrane of hydroxylate lecithin complex iodine/carboxymethyl chitosan/sodium alginate for burn wound healing using microwave-assisted drying. The composite membrane has good mechanical properties. Pan et al. [33] constructed a three-dimensional scaffold by using gelatin/SA composite solution blending with cells as the matrix materials. The structure was further stabilized using a Ca^2+^ cross-linking molding process, and the modified gelatin/SA showed the promise of successful cell adhesion [30,31,32]. The above studies all focused on the modification of sodium alginate and its biological characteristics, but there are relatively few studies on sodium alginate involving artificial skin. In addition, some researchers have studied CMC-Na. Alireza et al. [34] studied the physical properties of chitosan/carboxymethyl cellulose/starch biofilms as natural bio-based polymers, and the results showed that the presence of CMC-Na improved the tensile strength. Based on the properties of this material, we tried to use CMC-Na to modify the properties of SA, so as to improve the mechanical strength and provide the cell adhesion point of the scaffold.

Therefore, in this research, we used SA/CMC-Na blend hydrogels at different compositions than the printing precursors to fabricate the artificial skin in situ by using a 3D bio-printer. To obtain the optimum composition of SA/CMC-Na to fabricate the artificial skins, the blend hydrogel at different compositions was prepared. Following, the optimum composition of SA/CMC-Na was screened out by investigating the physical and chemical properties of SA/CMC-Na blending hydrogels. Finally, the optimum hydrogel precursor was used as the bio-ink to print the artificial skin for the rabbit acute skin wound repairing tests, and the autologous skin grafting was used as a control to illustrate the repair effect of the printed artificial skin. We believe this study is of great significance to the rational design and fabrication of SA/CMC-Na artificial skin for clinical application.

## 2. Materials and Methods

### 2.1. Materials

Used materials are as below: Sodium alginate (SA, medicinal grade, Qingdao Bright Moon Seaweed Group Co., Ltd., Qingdao, Shandong, China), sodium carboxymethyl cellulose (medicinal grade, Huzhou Zhanwang Pharmaceutical Co., Ltd., Huzhou, Zhejaing, China), anhydrous calcium chloride (CaCl_2_, analytically pure, Tianjin Hengxing Chemical Reagent Co. Ltd., Tianjin, China), anhydrous ethanol (analytically pure, Tianjin Tianli Chemical Reagent Co. Ltd., Tianjin, China), potassium bromide (analytically pure, Shanghai Alighting Biochemical Technology Co., Ltd., Shanghai, China), dimethyl sulfoxide (DMSO, biochemical reagent, Shanghai Alighting Biochemical Technology Co., Ltd., Shanghai, China), DMEM high-glucose medium, 3-(4,5 dimethythiazole-2)-2,5, diphenyltetrazolium bromide, fetal bovine serum (FBS), phosphate buffer (PBS), trypsin-EDTA, (Beijing Solarbio Technology Co. Ltd., Beijing, China), and mouse fibroblast cell line L929 (Wuhan Punoxai Life Technology Co., Ltd., Wuhan, Hubei, China). New Zealand rabbits were kept in a laboratory environment for 1 week, fed regularly daily, and observed. Following, the surgical experimental operations were performed on the New Zealand rabbits.

### 2.2. Preparation of SA/CMC-Na Hydrogel Precursors and the Artificial Skin Membrane

Using the ionized water as solvent, a series of SA/CMC-Na solutions with a mass percentage of 3 wt% were prepared. In these solutions, the composition ratios of SA and CMC-Na were 1:0, 1:1, 2:1, 4:1, 8:1, and 16:1, respectively. A constant-temperature magnetic agitator was used to stir the mixed solution at 80 °C for 6 h in a water bath. After SA and CMC-Na were completely dissolved, the mixed solution was taken out and let for 12 h to achieve the purpose of defoaming. Prior to the manufacturing process, 1 wt% CaCl_2_ solution was prepared with deionized water for reserve. Finally, the mixed solution of SA/CMC-Na as the hydrogel precursor was pneumatically deposited on a horizontal preparation platform by using a bio-printer pneumatic extrusion system (developed by our lab, see Appendix A). 

Figure 1 shows the artificial skin membrane fabricated by the printing technology, which possessed a two-tier structure. The diameter of the circular basilar membrane was 2.5 mm and its thickness was 2 mm. The side length of the reticular scaffold was 1.8 mm and its thickness was 1 mm. The composite scaffold material was printed on the petri dish. After the basilar membrane was printed, the bottom and side of the basilar membrane was semi-cross-linked with the atomizer for 10 s. It was ensured that the bottom of the scaffold could adhere to the surface of the scaffold’s upper layer or the petri dish effectively. Then, the next layer of scaffold (reticular scaffold) was printed and semi-cross-linked (the cross-linking method was consistent with that of the basement membrane) until the printing was complete. After the scaffold was printed, 1 wt% CaCl_2_ solution was added to the petri dish until the scaffold was completely immersed and fully cross-linked.

### 2.3. Characterization

#### 2.3.1. Morphology Analysis

Prior to the SEM observation, the fabricated membrane samples were freeze-dried by a lyophilizer (Shanghai Tianfeng TF-FD-27, Shanghai, China). One part of the samples was used to observe the surface morphology (the samples were first cut out of the cross section and then freeze-dried) and the other part of the samples was used to observe the cross-section morphology (the samples were first freeze-dried and then quickly cut out of the cross section). The fabricated membrane samples were coated with gold by an ion sputtering instrument (Hitachi MC1000, Japan) for 120 s to ensure their conductivity. The morphologies of the scaffolds were analyzed using SEM (Hitachi TM4000PLUS, Tokyo, Japan) at 15-kV voltage.

#### 2.3.2. Fourier Transform Infra-Red Spectroscopy (FTIR)

The membrane samples were put into an electric thermostatic blast drying oven at a temperature of 50 °C for 24 h. Following, the dehydrated samples were ground into powder and evenly mixed with KBr. The resulting mixed powder was used for FTIR analysis. The FTIR spectroscopy (BRUKER ALPHA II, Germany) scanning range was set for 4000–400 cm^−1^.

#### 2.3.3. Differential Scanning Calorimetry (DSC)

The DSC measurements were performed under a nitrogen atmosphere on a differential scanning calorimeter (Netzsch DSC214, Germany). All samples were dried in a vacuum oven for 24 h to remove moisture at 50 °C, and the crystallinity of the sample was analyzed by using a DSC (Netzsch DSC214, Germany) after it was restored to room temperature. After that, the samples were measured with a heating rate at 10 °C/min from 25 °C to 200 °C.

#### 2.3.4. Mechanical Property

The mechanical property of SA/CMC-Na hydrogel membranes were evaluated using an electronic universal material testing machine (Instron 5943, USA). The rectangular membranes, with size of 53 × 15 × 5 mm, were prepared for tensile testing, and the drawing speed was set as 20 mm/min. The cylindrical hydrogel samples, with size of Φ20 mm × 10 mm, were fabricated for compression testing. The press speed was set at 2 mm/min.

#### 2.3.5. Dehydration Rate Test

Each proportion of hydrogel membrane samples obtained by freeze-drying was soaked in distilled water, taking them out until they reached swelling equilibrium. We used the filter paper to remove all the excess moisture on the surface of each sample. After that, we weighed the samples and recorded the results. The samples with swelling equilibrium were dried in a vacuum drying oven for 24 h at 50 °C and the relative humidity was 50%. The weight of the hydrogel membrane was weighed regularly. The dehydration rate was calculated by the following equation:Dehydration rate = (*m_e_* − *m_b_*)/(*m_e_* − *m_a_*) × 100%(1)
where *m_e_* is the weight of the sample at swelling equilibrium, *m_b_* is the weight of the sample at different time points, and *m_a_* is the weight of the sample after 24 h.

#### 2.3.6. Degradation Rate Test

The degradation of the scaffold was assessed by immersing the cylinder sample (Φ20 mm × 10 mm) in a PBS/SBF buffer solution with pH of 7.4 at 37 °C and shaking continuously at 120 rpm on a vortex oscillator. These specimens were weighed after being removed from the degradation medium at a desirable time interval. The degradation property of the sample in vitro was quantitatively evaluated by calculating the weight loss rate of the sample in a dry state, which was calculated by the following equation:Degradation rate = (*m_o_* − *m_t_*)/*m_o_* × 100%(2)
where *m_o_* and *m_t_* are the weights of the tested sample before and after the degradation test, respectively.

### 2.4. Cytotoxicity Test

Scaffolds were immersed in FBS for 24 h at 37 °C in a CO_2_ incubator, and the extract liquid was sterilized for later use. L929 mouse fibroblasts at the logarithmic growth stage were taken and digested by trypsin-EDTA. In the following, the cell density was adjusted to 5 × 10^4^ cells per milliliter and the cells were inoculated in 96-well plates (the first column was used as a blank control without cell grafting). The cell suspension was 100 μL per well (5 × 10^3^ cells per well), cultured in the incubator for 24 h.

After the cells attached to the wall, the culture medium was absorbed and discarded. Then, 200 μL of complete culture medium was added to the first and second columns, respectively (blank in the first column and negative control in the second column), and 200 μL of complete culture medium solution containing different concentrations of materials was added to the remaining wells, respectively. The orifice plates were incubated in the incubator for 24 h and 48 h. Three multiple wells were set for each sample concentration to ensure the accuracy of the experiment.

After 24 h and 48 h, the waste liquid was absorbed and discarded, and serum-free medium and 20 μL 5 mg/mL-MTT were added to each well and incubated in an incubator for another 4 h. After that, the liquid in each well was discarded, and then 150 μL of DMSO were added to each well. After being shaken for 10 min, the optical density (OD) value of each well was detected at 490 nm (570 nm) of the enzyme marker.

### 2.5. Skin Repairing Experiments

New Zealand rabbits were used to make an acute, full-thickness, skin-defect model. A blank control group (B1) and two autologous skin transplantation control groups (B2 and B3) were established to verify the effectiveness of the composite hydrogel artificial skin with different compositions (A1, A2, and A3) in repairing skin defects. Prior to the skin repair experiments, 12 New Zealand rabbits weighing about 2.5 kg were raised in separate cages. Fasting was begun 12 h before the surgery. The operating room was disinfected by UV irradiation for 30 min before the operation. The rabbits were anesthetized by intravenous injection of 2% pentobarbital sodium at the ear edge at a dose of 40 mg/kg for anesthesia. Hair shearing, skin preparation, and disinfection were performed in the operating area, on the back. The limbs were fixed in the prone position, and the sterile hole towel was covered. Three full-thickness, skin-defect wounds with a diameter of 2.5 cm were made on both sides of the rabbit’s back at a distance of 3 cm from the spinal column. Each rabbit had six defect wounds, and the anterior and posterior defect wounds on the ipsilateral side were 3 cm apart. After the wound was cleaned, the samples were trimmed into the same diameter and tightly attached to the skin-defect surface. The autologous skin was sutured with surgery sutures. All the artificial skin membranes were prepared by the method described in Section 2.2. 

The sample marks were as follows: (1)Sample A1, the SC16:1 composite hydrogel membrane;(2)Sample A2, the SC8:1 composite hydrogel membrane;(3)Sample A3, the SC4:1 composite hydrogel membrane.

The designed experiments were as follows: (1)Gross appearance observation: At 1 week, 2 weeks, and 3 weeks after surgery, the overall skin healing, sample shedding, local redness and swelling, infection, and some other conditions were observed.(2)Sampling observation: At 1 week, 2 weeks, and 3 weeks after the operation, four rabbits were killed by air embolization. The repaired area and surrounding tissues were observed during sampling, such as the degree of vascular distribution, foreign body in cyst, etc.(3)HE histological observation: Based on organization of 4% formaldehyde solution, fixed, sliced paraffin embedding of 5 μm, the healing skin histology structure and normal skin residue sample variance were observed.

## 3. Results and Discussion

### 3.1. Physical and Chemical Properties’ Analysis

#### 3.1.1. Morphologies of SC Blend Hydrogels

As known, the miscibility of the two polymers can be reflected from the morphologies of their composites [35]. If the surface morphology of the binary polymer composite is smooth and uniform, the miscibility of the two polymers is good. On the contrary, if there are spherical particles or obvious separation interface on the surface, the miscibility of the two polymers is poor.

Without CMC-Na, the pure SA hydrogel membrane (Figure 2f) showed a relative flat cross section, which showed a good agreement with the smooth morphology feature of SA hydrogel. For the blend hydrogels of SC1:1 and SC2:1 (Figure 2a,b), there were many pellet particles and fibrous particles observed (Figure 2g,h), which means that at these compositions the SC blend hydrogels were incompatible. While with the mass ratio of SC went from 4:1 to 16:1 (Figure 2c–e), the pellet and fibrous particles were almost impossible to be found on the cross sections of the SC hydrogel membranes, which indicated that SA was completely compatible with CMC-Na at these compositions. That is to say, when the proportion of CMC-Na in SC was between 5.88% and 20%, SA and CMC-Na had a good miscibility.

On the other hand, the SEM analysis (in Figure 2a’–f’) revealed the influence of CMC-Na content on cross-section morphology of SC. Without CMC-Na, the pure SA membrane Figure 2f’ showed a lamellar pore structure similar to the structure of the CMC-Na-contained membranes at Figure 2a’,b’. Figure 2c’,d’ shows the faveolated porous structure. With the mass ratio of SC increasing from 1:1 to 16:1, the pore size in the cross section increased from tens of micrometers to hundreds of micrometers. The SC4:1 blend hydrogel membrane showed a proper size of pores, about 200–300 μm, for more effective cell adhesion [36,37,38,39]

#### 3.1.2. FTIR and DSC of Blend Hydrogels

If two materials are incompatible, there is no strong molecular interaction between the polymers of the two materials, and the infrared spectrogram of the composite is the superposition of the spectral bands of the two polymer components, as well the position width and relative strength of the peak not changing in the infrared spectrogram. On the contrary, when two materials are compatible, there will be a strong interaction existing between two polymers, and the resulting spectrum will be greatly deviated from that of several groups of single-component polymers, and there will be certain displacement and changes in the main absorption bands, as well as changes in the position width or relative strength of the peak [40].

From Figure 3a, for pure SA and pure CMC-Na, the broad peak at 3451 cm^−1^ was caused by the absorption of hydroxyl group (-OH) stretching vibration, but -OH stretching vibration peak of the SC blend hydrogels was 3423 cm^−1^. Obviously, the position of the peak changed. This change was due to the formation of hydrogen bonds between CMC-Na and SA. With the increase of CMC-Na content in SC, the peak width of the hydroxyl stretching vibration first increased and then decreased, and reached the maximum at SC4:1. It means that SA and CMC-Na had the strongest interaction at SC4:1. The absorption peaks near 1627 cm^−1^ and 1601 cm^−1^ were the carboxyl group (-COO-) antisymmetric stretching vibration peaks of SA and CMC-Na, while in the SC blend hydrogels, the position and intensity of -COO- antisymmetric stretching vibration peak also had a shift, and the intensity changes of SC4:1, SC8:1, and SC16:1 had little difference and decrease successively. The intensity changes of SC1:1 and SC2:1 were significantly weaker than those of the other groups. In addition, the absorption peak near 1417 cm^−1^ was -COO- symmetric stretching vibration peak and the absorption peak near 1324 cm^−1^ was -OH bending vibration peak for SA and CMC-Na, and the relative strength of the peak at 1417 cm^−1^ in CMC-NA was almost the same as that at 1324 cm^−1^. However, the vibration peak width of the SA/CMC blend hydrogel at 1307 cm^−1^ increased and the relative strength was higher than that at 1459 cm^−1^ (-COO- symmetric stretching vibration peak was shifted from 1417 cm^−1^). This was because -COO- and -OH in CMC and SA macromolecule formed hydrogen bonds, as well as the cross-linking effect of CaCl_2_. These factors limited the -OH bending vibration and enhanced the symmetric stretching vibration of -COO-. Similarly, with the increase of CMC-Na content in SC, the intensity change first increased and then decreased. It was obvious that the intensity of SC4:1, SC8:1, and SC16:1 was higher than that of SC1:1 and SC2:1, at 1459 cm^−1^, and SC4:1 had the greatest variation in intensity.

All of these mentioned above indicate that SA and CMC-Na were compatible when the composition ratio of SA/CMC-Na was at 4:1–16:1. This conclusion can also be reflected and verified by the DSC curves in Figure 3b, from which we can see that there were two melting point peaks in the DSC curves of SC1:1 and SC2:1 and only one melting point peak in the DSC curves of SC4:1, SC8:1, and SC16:1. One melting peak indicates a good miscibility, while more than one melting point peak indicates the immiscibility or partial miscibility or immiscibility of different polymers. That is to say, for the blend hydrogels of SC1:1 and SC2:1, the components in hydrogels were incompatible or partially incompatible, and for the blend hydrogels of SC4:1, SC8:1, and SC16:1, the components in hydrogels were compatible. This conclusion shows a good agreement with the morphology and FTIR analysis results.

#### 3.1.3. Mechanical Properties of Blend Hydrogel Scaffolds

The mechanical properties of SC blend hydrogel scaffolds are shown in Figure 3c,d. The calculated maximum load and the fracture stress are given in Table 1. By comparing the results, we know that the maximum load of SC4:1 and SC8:1 were greater than that of pure SA, and their tensile break stresses were more than 200 KPa, which is close to the tensile break stresses of mouse skin (240 KPa), rabbit skin (265 KPa), and human skin (150 KPa) [41,42,43]. Moreover, the nonlinear viscoelastic behavior of scaffolds was presented from their compression stress curves in Figure 3d. Herein, due to the high elasticity of the hydrogels, the compression was stopped when their displacement reached 6 mm (the total height of the sample was 10 mm), that is, when the deformation reached 60%. By comparing their compression stress curves, we know that, under the same deformation condition, the compression performance of SC4:1 was the best, followed by SC8:1, and both were stronger than that of pure SA.

#### 3.1.4. Dehydration Rate and Degradation Test of the Blend Hydrogel Scaffolds

The tested dehydration rate and degradation rate of pure SA and SC composite hydrogels are shown in Figure 3e,f, with the mass ratio of SC changed from 1:1 to 16:1. It exhibited a rapid slow rate of the dehydration (Figure 3e). Some studies have shown that a moist environment can help wound healing [44]. The dehydration rate test studies the moisturizing ability of the blend hydrogel scaffolds, which have a certain influence on keeping the wound moist. Therefore, it is of great significance to study this performance. As can be seen from Figure 3e, for pure SA and SC composite hydrogels, the variation trend of the dehydration rate was similar. As time went on, the dehydration rate of each sample gradually increased. In the first 8 h, the dehydration rate of the hydrogel rose sharply and tended to be stable at around 24 h. The dehydration rate varied with the content of CMC-Na in SC. Among them, the moisturizing ability was sorted as SC4:1 > SC8:1 > SA > SC16:1 > SC2:1 > SC1:1. This is because the miscibility of SA and CMC-Na blends was different, and the cross-linking degree of polymer chains was also different, leading to different hydrogen bonding. Finally, these factors affected the rate of water loss in hydrogels. The blends of SA and CMC-NA were explained in the previous section, and the experiment in this section also showed that SC4:1 had a good moisturizing ability. Additionally, it also showed a slightly accelerated degradation rate during first 7 days (Figure 3f). After 15 days, the degradation rate of each component gradually tended to be stable, which was conducive to the replacement of composite materials in the degradation process. In the wound healing of skin tissue, the basic process of wound healing contains an acute inflammation stage, cell proliferation stage, and scar formation stage. SC4:1, SC8:1, and SC16:1 performed a relatively appropriate degradation rate, maintaining a smaller degradation rate in the first 2 weeks, which was beneficial for cell proliferation during this time. By contrast, with granulation tissue and subsequent scar tissue formation, a relatively large degradation rate was needed to provide enough space, corresponding with the degradation rates of SC1:1, SC2:1, and SC4:1.

### 3.2. In Vitro Cytotoxicity Test

This study employed an in vitro co-culture method characterized by static co-incubation of L929 mouse fibroblasts with SC composite hydrogel scaffolds’ extract liquid. The results are shown by the statistical data in Table 2. The relative cell proliferation rates in each group were all above 95%, and the component of hydrogel had little effect on the proliferation rate of cells. Compared with the toxicity standard, the toxicity of SC composite hydrogel scaffold was at level 1 [45], which indicates the prepared SC composite hydrogels possessed a good biocompatibility. That is to say, the SC composite hydrogel has the potential to be used as an implant material without any toxic side effects on cells.

Therefore, by comprehensively considering the morphological structure, thermal properties, mechanical properties, and biological properties of the SC composite hydrogels at different compositions, it was concluded that the composite hydrogel of SC4:1 and SC8:1 have relatively good comprehensive properties on the premise of good miscibility between SA and CMC-Na.

### 3.3. Skin Repairing Experiments

To further investigate the effect of the composition ratio on the skin repairing, here SC16:1, SC8:1, and SC4:1 blend hydrogels were used as the matrix bio-ink to fabricate the artificial skin for the skin defeat repairing. Figure 4 shows the operation procedures and treatment positions.

For the SC skin scaffold groups (A1, A2, and A3), after the first week (Figure 5), all wounds shrunk significantly, with a diameter of about 1.5 cm. Most of the hydrogel samples at positions A1, A2, and A3 were still sticking to the wound surface, which is similar to “skin crust” with a dark, yellowish-brown color and dry surface. After 2 weeks, the diameter of scar tissue reduced to about 1 cm with varying degrees of scab. After 3 weeks, the scars shrunk to about 0.5 cm, while the color of all wounds was closer to the normal skin than before, the shape of the scars was irregular, and back hairs grew more than half locally.

For the control groups (B1, B2, and B3), after the first week, there was no filling in the wound at the blank control of B1, but it showed obvious shrinkage and a bright red, sunken shape with some scabs with a diameter of about 0.8~1.5 cm (1W-1, 1W-2 in Figure 5). Only one rabbit (see Figure 5 1W-4) showed brown skin at the B3 site. After 2 weeks, the scar area shrunk to 0.8 cm diameter and the healing surface was uneven (2W-1, 2W-4 in Figure 5). After 3 weeks, scar contraction was more obvious (3W-1, 3W-2 and 3W-3 in Figure 5), the healing surface was not smooth, and the surrounding normal skin was still tight. For the autologous skin controls of B2 and B3, after the first week, the grafted skin healed well with a ruddy skin color, and most of the sutures were intact with hair grown locally. After 2 weeks, the skin sheet fully survived without obvious shrinkage, the color was ruddy and whitish, and O-shaped healing lines were still visible locally at the junction of the skin sheet and normal skin. The damaged parts of the epidermis at B2 and B3 were bright red due to the animal’s scratching. After 3 weeks, O-shaped scars were as small as 0.5 cm, the surrounding skin looked moderately elastic, and the back hair grew more than half.

According to the general observation, from the first week to the third week, the wound surface of the sample group contracted significantly. The diameter of the wound surface changed from 2.5 cm to about 0.8 cm. The shrinkage rate was basically consistent with the degradation rate of the composite material in vitro. Compared with the blank group, the wound shrinkage was less obvious. This can be attributed to the SC4:1 hydrogel skin scaffold having a certain mechanical strength. In the process of skin tissue repair, the SC4:1 hydrogel skin scaffold could be used to fill the wound defect and play a certain supporting role to prevent excessive skin contraction during the formation of scar tissue in wound healing. In organisms, scar contraction often leads to organ deformations and even dysfunction, and the formation of scar tissue is a gradual fibrosis of granulation tissue during healing. In this process, blood vessels are sparse, the number of cells decrease sharply, and capillaries close, which is not conducive to the formation of skin accessory organs, such as hair follicles and blood vessels. The autologous skin control groups showed that the autologous skin was almost not contractile, which can be attributed to the non-absorbable suture used for sewing, while the hydrogel skin scaffolds groups were only fixed by gauze and medical tape. During wound healing, most of the samples were attached to the wound surface, indicating that there were cells and new tissue on the scaffold structure. It showed a possible way to fix the artificial skin with customized structure on the wound sites, avoiding pain during a skin grafting suture operation.

### 3.4. Histological Analyses

Histological examination (HE) observation of skin wound repair is shown in Figure 6. After the first week, for the SC sample groups A1, A2, and A3, the healing areas were filled with granulation tissue. The surface was uneven with breaks in A2, and the local epidermal layer at A2 and A3 began to form. New capillaries were dense at the junction with normal tissue. For the blank control group B1, the healing surface was significantly wider than that of SC sample groups, the skin scabs closely adhered to the surface of the healing area, and the formation of the epidermal layer was not obvious. For autologous skin control groups B2 and B3, there was no obvious difference between the connective tissue of the dermis and the normal tissue, and a small number of sweat glands and hair follicles were formed in the dermis.

After 2 weeks, structures such as sweat glands, hair follicles, and small blood vessels appeared in the healing tissue (A3 at 2 weeks in Figure 6), which was not obvious in A1 or A2. The healing surface was shorter than that at 1 week, and the junction with the normal tissue dermis was more obvious. The epidermis at A2 and A3 was basically formed, and the epidermis at A1 was still crawling. For B1, the healing area was filled with granulation tissue without skin accessory structure. The healing surface was relatively wider than that of the A1, A2, and A3 groups, but the epidermal layer was basically formed. For autologous skin control groups B2 and B3, the boundary between the epidermal layer of subcutaneous connective tissue and normal tissue was still not obvious. The number of sweat glands, hair follicles, sebaceous glands, small blood vessels, and hair shafts in the local dermis was less than normal tissues. 

After 3 weeks, the healing surface of each position was significantly shortened, the junction of the dermis and normal tissues at A1, A2, A3, and B1 became more obvious, and the epidermis formed well. Noticeably, there were many structures such as sweat glands, hair follicles, sebaceous glands, and small blood vessels in the dermis at A3, which looked close to normal tissues. A2 had a small number of these accessory organs (sweat glands, hair follicles, sebaceous glands, and small blood vessels), and A1 had fewer accessory organs. Autologous skin control groups B2 and B3 showed all normal skin structures with sweat glands, hair follicles, and sebaceous glands.

By histological observation at the third week, the dermis of the A3 group showed a number of sweat glands, hair follicles, sebaceous glands, small blood vessels, and other structures, which were very close to the normal tissues. However, A2 had a small number of these accessory organs (sweat glands, hair follicles, sebaceous glands, and small blood vessels), and A1 had fewer ancillary organs than A2. This indicates that the recovery effect of SC4:1 hydrogel multilayer reticular skin was superior to those of SC16:1 and SC8:1 hydrogel multilayer reticular skin. In addition, SC4:1 hydrogel multilayer reticular skin can achieve the recovery effect of autologous skin grafting when it is used to repair wounds. Therefore, combining the characterized properties above, we know that SC 4:1 is the most appropriate one to fabricate the artificial skin for wound repairing.

## 4. Conclusions

In this study, SA/CMC-NA blending hydrogels were proposed to fabricate the artificial skin membrane for wound repair. To screen out an appropriate SA/CMC-NA blend for skin scaffold fabrication, firstly, the blend hydrogels at different compositions were designed and characterized. SEM was used to analyze the surface morphology. FTIR was used to reveal the changes of functional groups in the SA/CMC-NA blend hydrogel. A DSC experiment was used to measure the melting point peak of SA/CMC-NA composites. These experiments explored the miscibility of the two materials, when the proportion of CMC-Na in blend hydrogel was 33%–50%, the blend system showed immiscibility. When the proportion of CMC-Na in blend hydrogel was less than or equal to 20%, the miscibility of the two materials was good. Moreover, the results of morphological structure, thermal properties, mechanical properties, and biological properties of the SC composite hydrogels at different compositions showed that the blend hydrogels of SC4:1 and SC8:1 had relatively good comprehensive properties. Finally, the skin scaffolds with a two-tier structure were printed by using SC16:1, SC8:1, and SC4:1 blend hydrogels as the matrix, and then used for rabbit acute skin wound repair tests. Results show that SC4:1 blend hydrogel can achieve a recovery effect similar to autologous skin grafting. This research is of great significance for the implantable tissue-engineered skin scaffold, which provides a possibility and basis for the repair of large-area skin defect.

## Figures and Tables

**Figure 1 gels-07-00115-f001:**
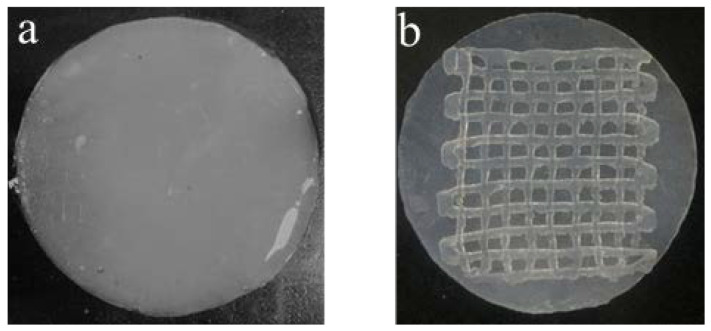
Appearance of SA/CMC-Na hydrogel membranes fabricated by our pneumatic extrusion printing system, (**a**) one layer as a basilar membrane and (**b**) three layers as a scaffold (two layers reticular scaffold and 1onelayer basilar membrane).

**Figure 2 gels-07-00115-f002:**
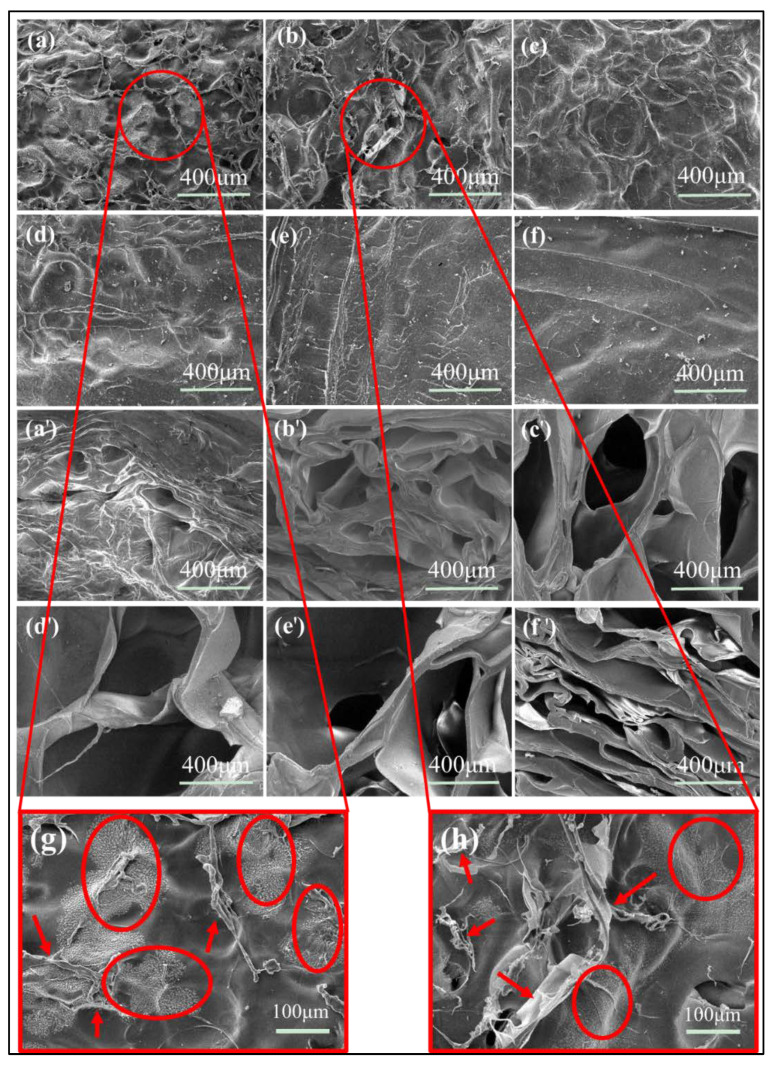
SEM images of SC composite membranes with different mass ratios at surface versions: (**a**) SC1:1; (**b**) SC2:1; (**c**) SC4:1; (**d**) SC8:1; (**e**) SC16:1; and (**f**) SA. SEM images of SC composite membranes with different mass ratios at cross-section versions: (**a’**) SC1:1; (**b’**) SC2:1; (**c’**) SC4:1; (**d’**) SC8:1; (**e’**) SC16:1; and (**f‘**) SA. Observation surface morphology of composite scaffold with two ratios (×200), (**g**) SC1:1, (**h**) SC2:1.

**Figure 3 gels-07-00115-f003:**
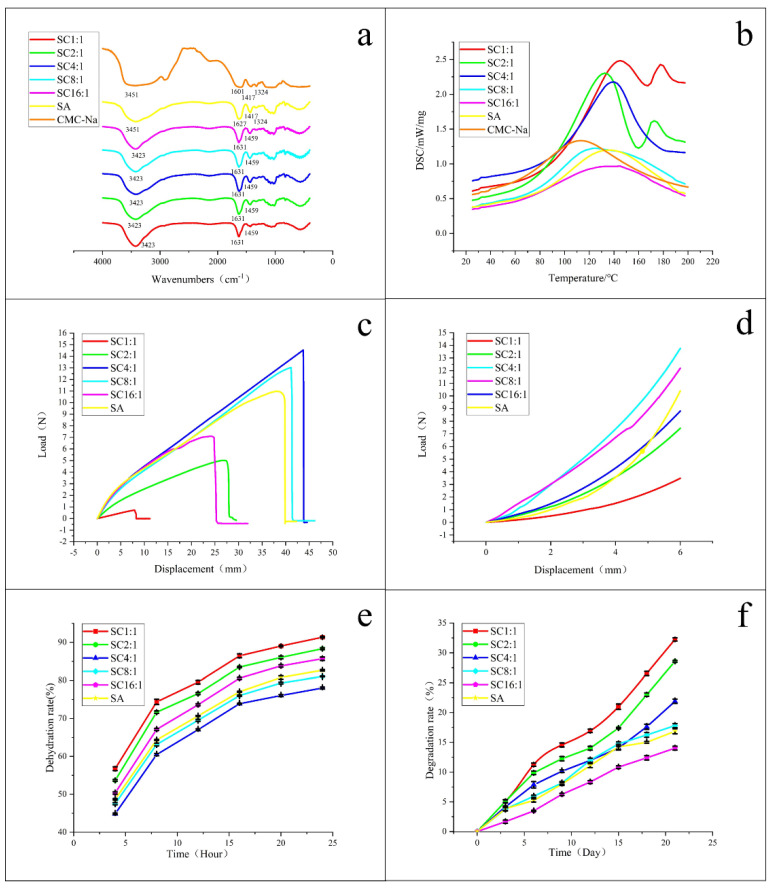
(**a**) FT-IR and (**b**) DSC curves of SC composite hydrogels; (**c**) tensile performance and (**d**) compression stress of support when compression deformation was 60% of SC composite scaffolds with different mass ratios; (**e**) solution viscosity test and (**f**) in vitro degradation rate of SC composite scaffolds with different mass ratios (each value represents the mean ± S.D. (*n* = 3)).

**Figure 4 gels-07-00115-f004:**
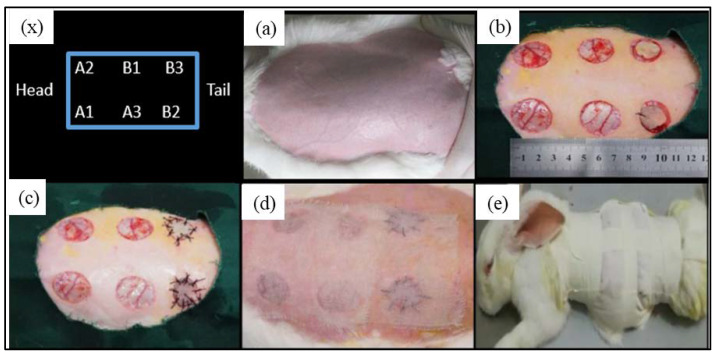
(**a**–**e**) The operation procedure of skin wound repairing; x is the schematic diagram of sample placement position: A1 refers to SC4:1 hydrogel skin membrane; A2 and A3 refer to SC4:1 with growth factor; B1 is a blank control; B2 and B3 are the autologous skin controls.

**Figure 5 gels-07-00115-f005:**
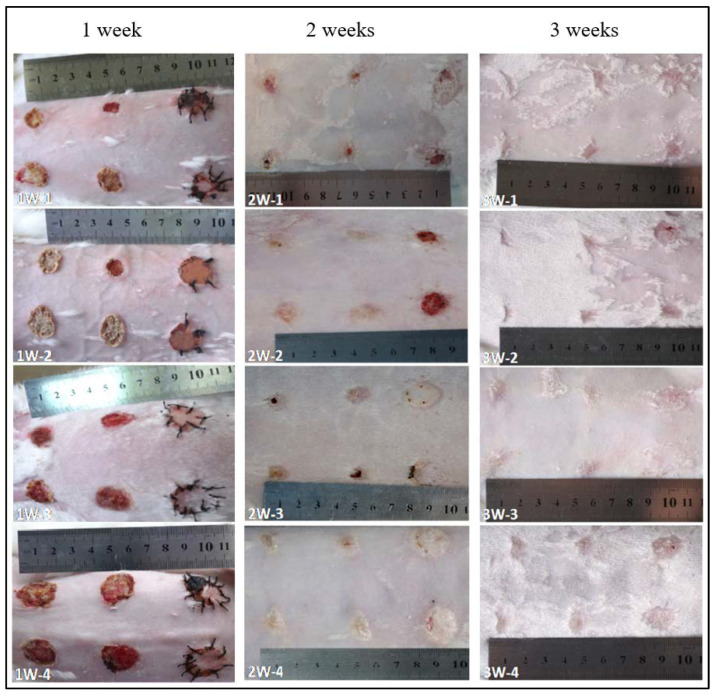
The photographs of skin wound repair at 1 week, 2 weeks, and 3 weeks.

**Figure 6 gels-07-00115-f006:**
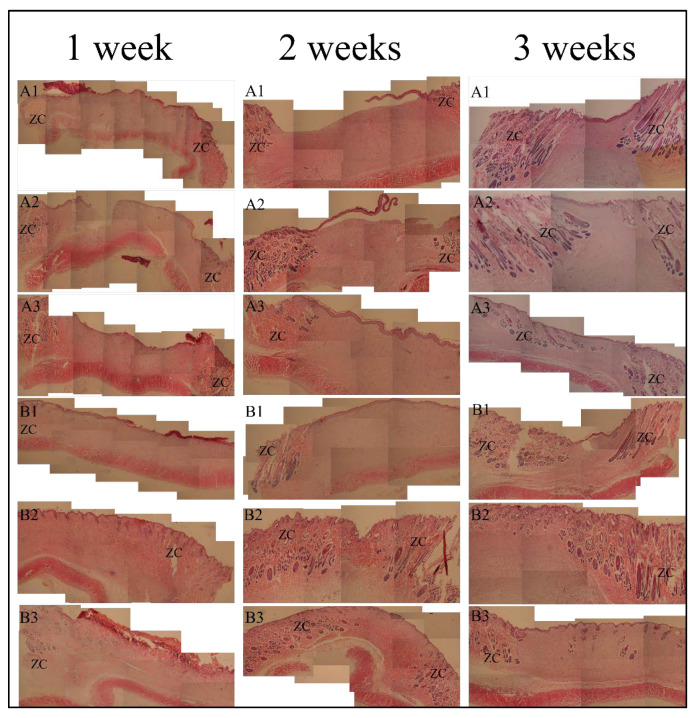
The observation results of HE at 1 week, 2 weeks, and 3 weeks (×40, ZC stands for normal tissue). A1 is the SC16:1 composite hydrogel membrane; A2 is the SC8:1 composite hydrogel membrane; A3 is the SC4:1 composite hydrogel membrane; B1 is a blank control group; B2 and B3 are two autologous skin transplantation control groups.

**Table 1 gels-07-00115-t001:** The tensile break stress and compression stress of SC blend hydrogels at different compositions. Each value represents the mean ± S.D. (*n* = 3).

Sample	Mass Ratio (SA:CMC-Na)	Maximum Tensile Load [N]	Tensile Break Stress [KPa]	Compressive Strain Value [%]	Compressive Load [N]	Compressive Stress [KPa]
SC1:1	1:1	0.722 ± 0.018	12.037 ± 0.293	20	0.512 ± 0.006	1.630 ± 0.020
				40	1.512 ± 0.022	4.812 ± 0.070
				60	3.488 ± 0.173	11.103 ± 0.550
SC2:1	2:1	5.014 ± 0.127	83.568 ± 2.121	20	1.245 ± 0.068	3.964 ± 0.218
				40	3.584 ± 0.052	11.408 ± 0.167
				60	7.444 ± 0.316	23.696 ± 1.007
SC4:1	4:1	14.531 ± 0.617	242.190 ± 10.287	20	2.982 ± 0.097	9.491 ± 0.309
				40	7.385 ± 0.182	23.506 ± 0.580
				60	13.765 ± 0.662	43.816 ± 2.106
SC8:1	8:1	13.016 ± 0.379	216.937 ± 6.313	20	3.025 ± 0.093	9.630 ± 0.297
				40	6.715 ± 0.275	21.375 ± 0.876
				60	12.184 ± 0.264	38.782 ± 0.842
SC16:1	16:1	7.099 ± 0.301	118.311 ± 5.018	20	1.475 ± 0.091	4.695 ± 0.291
				40	4.301 ± 0.159	13.692 ± 0.505
				60	8.804 ± 0.225	28.023 ± 0.717
SA	1:0	10.976 ± 0.278	182.938 ± 4.628	20	1.002 ± 0.045	3.190 ± 0.144
				40	3.572 ± 0.160	11.370 ± 0.508
				60	10.398 ± 0.522	33.098 ± 1.478

**Table 2 gels-07-00115-t002:** Average OD values and RGR values at different time points in each group. Each value represents the mean ± S.D. (*n* = 3).

Group	Time	The Average OD Value	RGR (%)
SC1:1	24 h	0.215252 ± 0.002491	97.8653
		0.00249054	
		0.00249054	
	48 h	0.302968 ± 0.003605	97.9326
SC2:1	24 h	0.214738 ± 0.002640	97.5381
	48 h	0.303181 ± 0.003210	98.0242
SC4:1	24 h	0.215666 ± 0.002373	98.1293
	48 h	0.303576 ± 0.002564	98.1947
SC8:1	24 h	0.214223 ± 0.002073	97.2101
	48 h	0.302142 ± 0.004031	97.5766
SC16:1	24 h	0.212932 ± 0.003814	96.3877
	48 h	0.299932 ± 0.005959	96.6239
SA	24 h	0.212497 ± 0.003442	96.1106
	48 h	0.296801 ± 0.005505	95.2741
Negative control	24 h	0.218603 ± 0.002600	
	48 h	0.307795 ± 0.003892	
Blank control	24 h	0.061614 ± 0.001829	
	48 h	0.075782 ± 0.000688	

## Data Availability

Data is contained within the article and Appendix A.

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
