# Peer review of "Design and Fabrication of Sodium Alginate/Carboxymethyl Cellulose Sodium Blend Hydrogel for Artificial Skin"

_gels, 2021, doi:10.3390/gels7030115_

Round 1
Reviewer 1 Report
The presented paper "Design and fabrication of sodium alginate/carboxymethyl cellulose sodium blending hydrogel for artificial skin" is interesting and relevant to the fields of biomaterials and tissue engineering. The article presents relevant results with skin scaffold assays (SA/CMC-NA blending hydrogels) in rabbits, it can be accepted to consider some points
*Some work is necessary spell check required.
* correct the subscript to wavenumber (cm-1).
*Figure 3 does not show good quality. Mainly the FTIR figure.
* it would be interesting to show in the supplementary material the 3d printing system
Reviewer 2 Report
The authors prepared a range of sodium alginate - carboxymethyl cellulose blends to use as materials for wound regeneration. The authors characterized the physical and chemical properties of the blends and tested their biocompatibility in vitro (exposing murine fibroblasts to material extracts) and in vivo (treating skin deletions on mice using 3D printed patches).
The authors should showcase that all ethical requirements in terms of animal experiments were fulfilled.
Sentence structure is often strange and should be revised.
line 65 - what does 'high efficiency' mean in terms of using sodium alginate as a material?
line 69 - what are 'poor mechanical properties'?
Preparation of SA/CMC hydrogels:
- How were the materials sterilized? Or was heating to 80°C considered sufficient?
- Which growth factors were used (line 131)?
- The cross-linking using an atomizer requires further description. Also, why was the first layer cross-linked separately? In this reviewer's experience, this can reduce the binding of successive layers (especially with alginate).
Characterization:
- It should be noted that SEM analysis is performed on dry samples, which are very different from wet samples used in practice.
- What is the purpose of the scanning calorimetry analysis.
- How were the samples mounted for the tensile tests to ensure repeatable measurements?
- I don't quite understand the degradation procedure. Why were the samples vortexed. How does this mimic the native conditions. How could a 'constant weight' be achieved? I would expect the samples to degrade until they dissolved completely.
Cytotoxicity test:
- How were the extracts sterilized?
- This section is difficult to understand, please rephrase the description.
Skin repairing experiments:
This section is also difficult to understant, please rephrase the description.
Morphologies of the SC blend hydrogels:
- What does compatibility of SA and CMC mean?
- The lyophylization process can have a major impact on water crystal formation and consequently the porosity and morphology of the scaffolds. How can the authors be sure, that the results are not a consequence of that instead of the SA-CMC ratio?
Mechanical properties of the hydrogels scaffolds:
- Compressive load and stress without distance/strain values is meaningless. Table 1 needs to be adjusted.
Skin repairing experiments:
- The authors state that hydrogels in SC ratios 16:1, 8:1 and 4:1 were used for the sind regeneration experiments. However, only results of 4:1 experiments are shown.
- Figure is difficult to asses. Please provide larger images and a more comprehensive description in the caption.
